# Protocol for chronic hepatitis B virus infection mouse model development by patient-derived orthotopic xenografts

**Aleksey M. Nagornykh** [ORCID]ᵒ*, **Marina A. Tyumentseva**ᵒ, **Aleksandr I. Tyumentsev**ᵒ, **Vasiliy G. Akimkin**

Central Research Institute for Epidemiology of the Federal Service for Supervision of Consumer Rights Protection and Human Welfare, Moscow, Russia

ᵒ These authors contributed equally to this work.
* nagornih@cmd.su

**Funding:** This work is supported by the Ministry of Science and Higher Education of the Russian Federation within the framework of a grant in the form of a subsidy for the creation and development

## Abstract

### Background

According to the World Health Organization, more than 250 million people worldwide are chronically infected with the hepatitis B virus, and almost 800.000 patients die annually of mediated liver disorders. Therefore, adequate biological test systems are needed that could fully simulate the course of chronic hepatitis B virus infection, including in patients with hepatocellular carcinoma.

### Methods

In this study, we will assess the effectiveness of existing protocols for isolation and cultivation of primary cells derived from patients with hepatocellular carcinoma in terms of the yield of viable cells and their ability to replicate the hepatitis B virus using isolation and cultivation methods for adhesive primary cells, flow cytometry and quantitative polymerase chain reaction. Another part of our study will be devoted to evaluating the effectiveness of hepatocellular carcinoma grafting methods to obtain patient-derived heterotopic and orthotopic xenograft mouse avatars using animal X-ray irradiation and surgery procedures and *in vivo* fluorescent signals visualization and measurements. Our study will be completed by histological methods.

### Discussion

This will be the first extensive comparative study of the main modern methods and protocols for isolation and cultivation primary hepatocellular carcinoma cells and tumor engraftment to the mice. All protocols will be optimized and characterized using the: (1) efficiency of the method for isolation cells from removed hepatocellular carcinoma in terms of their quantity and viability; (2) efficiency of the primary cell cultivation protocol in terms of the rate of monolayer formation and hepatitis B virus replication; (3) efficiency of the grafting method in terms of the growth rate and the possibility of hepatitis B virus persistence and

of the «World-class Genomic Research Center for Ensuring Biological Safety and Technological Independence under the Federal Scientific and Technical Program for the Development of Genetic Technologies», agreement No. 075-15-2019-1666. The funders had and will not have a role in study design, data collection and analysis, decision to publish, or preparation of the manuscript.

**Competing interests:** The authors have declared that no competing interests exist.

replication in mice. The most effective methods will be recommended for use in translational biomedical research.

## Introduction

Chronic hepatitis B virus (HBV) infection is characterized by the persistence of surface antigen (HBsAg) for at least six months (with or without the simultaneous presence of HB viral protein (HBeAg)). The continuous presence of HBsAg is the main marker of the risk of developing chronic liver disease and hepatocellular carcinoma (HCC) during the life [1]. Chronic inflammation characterized by repeated cycles of apoptosis, necrosis, and regeneration is an important contributor to hepatocarcinogenesis [2].

To date, there is a huge variety of animal models for studying HBV infection in the world, but mouse models are still the most popular and cost-effective [3]. For all its diversity, mouse models have their advantages and disadvantages, depending on the purpose. Thus, one of the first mouse models of HBV infection, HBV-Trimera mice, was created by transplanting HBV-infected liver tissues under a kidney capsule of mice with induced severe combined immunodeficiency [4]. Unfortunately, this model is not suitable for development of immunotherapy strategies or adaptive immunity studies, since implanted human hepatocytes remain functional just for 1 month allowing only short-term anti-HBV therapies [5]. Delivery of HBV replicons or the HBV genome associated with adenovirus carriers by hydrodynamic injection into the tail vein of mice remains perhaps the most common way to simulate HBV infection [6–10]. However, only temporary replication of HBV is observed—the peak of viremia is reached on the 6th day after injection, followed by a rapid decline [11]. In addition, there is no direct infection of mice with HBV, and over time, HBV-specific antiviral antibodies and cytotoxic T-lymphocytes appear.

The development of humanized mice made it possible to keep the long-term replication and persistence of HBV in animal models. Transgenic mice with albumin-urokinase-type plasminogen activator (*uPA*) were the first of them to successfully demonstrate repopulation of hepatocytes of a healthy adult human to the diseased mouse liver. To improve the recovery of xenograft hepatocytes, some researchers have crossed these mice with immunodeficient mice (SCID). Thus, uPA-SCID transgenic mice were derived. However, unpredictability in maintenance and high cost of mice prevents the widespread use of this model [12, 13].

Generation of mice deficient in fumarylacetoacetate hydrolase (*FAH*) gene, an enzyme that plays an important role in the last steps of tyrosine catabolism pathway. Damage to the hepatocytes of Fah KO mice occurs due to the accumulation of toxic intermediate products of tyrosine metabolism. The severity of liver damage is regulated by the introduction of 2-(2-nitro-4-fluoromethylbenzoyl)-1,3-cyclohexanedione into the mice. To prevent rejection of human hepatocytes, Fah KO mice were crossed with *Rag2/IL2rg* KO mice, thus generating mice with a triple knockout of genes, called FRG KO mice. *Azuma et al* describe the steady expansion of human hepatocytes and the maintenance of high production of HBV in serum of FRG KO mice [14]. A significant limitation in the widespread use of FRG KO mice, apparently, is their high cost.

Similarly, TK-NOG mice were briefly exposed to the drug ganciclovir also maintain a high possibility of HBV replication [15, 16]. However, the demand for this mouse model is not so high compared to FRG KO mice due to male infertility, which ultimately leads to low reproduction efficiency [17].

Unfortunately, most of these immunodeficient chimeric human liver mice do not have a functional immune system, which makes it impossible to study human-specific therapeutic strategies and immune responses caused by HBV. In an attempt to overcome these limitations, mice with double humanization are generated. Thus, AFC8-hu HSC/Hep mice are mainly used for HCV researches [18, 19], A2/NSG/Fas-hu-HSC/Hep mice maintain the persistence of HBV and are susceptible to the development of chronic hepatitis and liver fibrosis [20], After transplantation of an adult human hepatocytes into uPA-NOG transgenic mice, about 70% of the mouse liver is humanized [17]. HBV-infected HIS-HUHEP mice are able to maintain a high level of viremia and demonstrate a phenotype of chronic inflammation, and hBMSC-FRGS mice demonstrate the development of acute or chronic hepatitis with varying degrees of lymphocytic portal inflammation, liver fibrosis progressing to cirrhosis, but the development of HCC does not occur, which may be due to the duration of the pathogenesis of HCC in humans [21]. With regard to FRGN mice, their use as HBV models is allowed in the future [22].

Chronic HBV infection throughout life usually leads to chronic hepatitis, fibrosis, cirrhosis of the liver or hepatocellular carcinoma HBV accounts for about 50% of all HCC etiologies [23]. HCC is the second leading cause of cancer death in male, and the sixth leading cause in female around the world [24].

HBV can contribute to carcinogenesis by three different mechanisms: (1) classical retrovirus-like insertion mutagenesis with the integration of viral DNA into host cancer genes, such as *TERT*, *CCNE1* and *MLL4*; (2) stimulation of genomic instability as a result of both the integration of viral DNA into the host genome and the activity of viral proteins; (3) the ability of wild-type and mutated/truncated viral proteins (HBx, HBc and preS) to influence cell functions, activate oncogenic pathways and sensitize liver cells to mutagens [25, 26]. The integration of HBV DNA into the host genome causes genomic instability and direct insertion mutagenesis of various cancer-related genes. A clonal expansion of hepatocytes containing unique virus-cell DNA junctions formed by the integration of HBV DNA can be detected in patients at various stages of chronic infection. Unfortunately, the nature of the selective advantages that sustain the expansion of hepatocyte clones containing integrated HBV DNA and whether they represent a true pre-neoplastic condition as well as their relation with early HCCs are still unclear [25].

Developed animal models are based on various strategies for inducing liver neoplasms in mice: (1) genetically engineered models expressing specific fragments of the HBV and HCV genome [27, 28]; (2) models obtained with the help of chemical toxic agents (carcinogens) by directly damaging DNA or contributing to the formation of a tumor after administration with a hepatotoxic compound that promotes the spread of preneoplastic cells [29]; (3) the use of humanized mice [30–32]; 4) transplantation of immortalized cell lines [33–35].

In recent years, models based on the transplantation of fragments of organs or tissues obtained from patients have become popular [36, 37]. Many of them are obtained from patients with aggressive oncological diseases, including liver cancers. Many techniques have already been proposed for the isolation, cultivation and transplantation of primary cells obtained from patient biopsies, which are often very different [38, 39].

The currently existing *in vitro* and *in vivo* models of viral diseases make it possible to observe morphological changes in target organs of living animals through bioluminescence and fluorescent intensity imaging during the development of the pathological picture of the disease, thus allowing to assess the persistence and replication of viral DNA in the body of the animal model, noting their gradation over time, using methods that do not require periodic euthanasia of the animal [40–44]. However, the protocols differ in this direction as well.

Therefore, we plan to conduct series of experiments on choosing the optimal protocol for creating a patient-derived orthotopic xenograft (PDOX) mouse model for chronic HBV infection studying and xenograft visualization techniques using fluorescent imaging.

## Materials and methods

### Study aim

The aim of the study is to evaluate the efficiency of protocols for isolation and cultivation of primary cells from diseased liver tissue included primary HCC cells; to optimize methods for creating HCC-HBV-PDOX avatars and compare the obtained data with the results of hematological, biochemical and molecular tests of the patient-derived biomaterials.

### Study objectives

1. To estimate the efficiency of primary HCC cells isolation using primary HCC cells isolation protocols.

2. To evaluate the efficiency of primary HCC cells cultivation using primary HCC cells cultivation protocols.

3. To give an assessment of the efficiency of HCC-HBV-PDOX avatars development protocols.

4. To analyze the correlation between biochemical blood analysis and complete blood count parameters, HBsAg, HBV DNA and AFP quantity of graft donors with *in vitro* and *in vivo* characteristics of these grafts.

### Study design and settings

We will conduct a comparative study of various methods of isolation and cultivation of primary liver cells and optimize these protocols for primary HCC cells (Fig 1). We will implement a comparative study of several methods of HCC grafting to mice and evaluate the total mouse

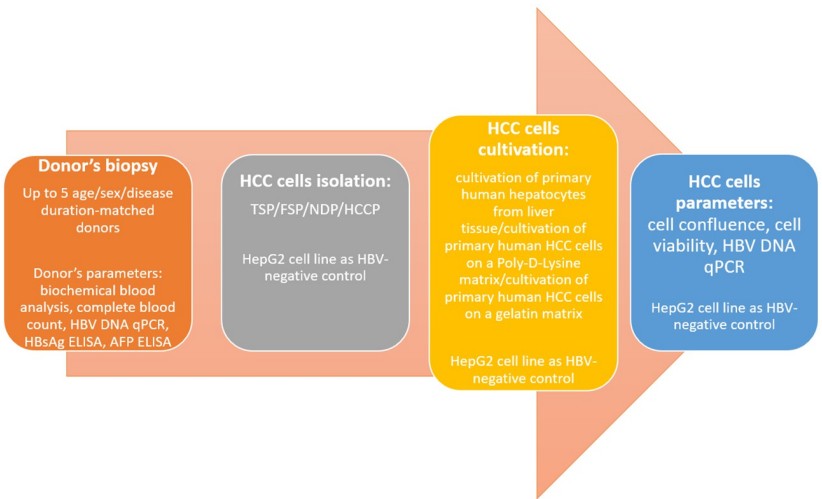

**Fig 1. Graphics pipeline of HCC cell protocols research.**

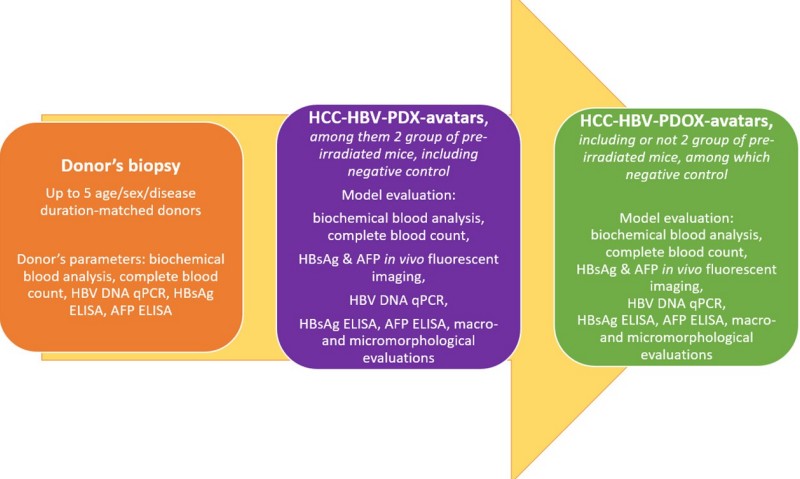

**Fig 2. Graphics pipeline of animal protocols research.**

body X-ray pre-irradiation effect on xenograft growth and *in vivo* HBV DNA replication (Fig 2).

## Study methods

The study will be conducted using the following methods:

1. primary cells isolation from biopsy material;

2. cell viability assessment;

3. cells cultivation;

4. qPCR;

5. flow cytometry (immunocytochemistry);

6. surgical manipulations with laboratory animals;

7. *in vivo* fluorescent intensity imaging;

8. pathomorphological evaluations of HCC-HBV-PDOX avatars;

9. immunohistochemical (IHC) analysis.

## Study outcomes

After the study completion we will possess the data regarding: (1) efficiency of primary HCC cells isolation and cultivation protocols; (2) efficiency of the methods for transplantation of patient-derived xenograft to mice; (3) correlation data between the donor's blood parameters and animal models; (4) *in vivo* HBsAg and AFP fluorescent intensity imaging. The methods used will include commercially available animals, immortalized cell line, reagents and materials.

We expect that a comparative characterization of biochemical blood analysis and complete blood count parameters, HBsAg, HBV DNA and AFP quantity of graft donors with *in vitro* and *in vivo* characteristics of these grafts will allow us to obtain a correlation for the

replicability assessment of chronic HBV infection combined with HCC in mouse model. This correlation can be used in translational biomedical research of chronic HBV infection combined with HCC.

We will conduct the first large-scale study of AFP and HBsAg distribution and localization *in vivo*, performed by the small animal optical imaging system Newton 7.0 FT-500 and confirmed by images. Also, we will provide optimized protocols for primary HCC cells isolation and cultivation.

## Study details

All procedures with patients will be performed in accordance with the World Medical Association Declaration of Helsinki Ethical Principles for Medical Research Involving Human Subjects [45]. An informed consent on voluntary participation in the study will be signed with each biomaterial donor. HCC biopsy material will be obtained from up to five patients of one sex, age, duration and severity of the disease and confirmed diagnosis of "HBV-induced HCC".

The results of clinical tests obtained from patients during 6 months of observation will be analyzed according to the following indicators:

- biochemical blood analysis—alanine aminotransferase (ALT), aspartate aminotransferase (AST), total bilirubin (TB), conjugated bilirubin (CB), alkaline phosphatase (AP), gamma-glutamyl transferase (GGT), cholesterol (Chol);

- complete blood count;

- HBV DNA quantitative polymerase chain reaction (qPCR);

- HBsAg quantitative ELISA;

- AFP quantitative ELISA.

*1. Primary HCC cells isolation*

1.1 Two-stage protocol for isolation of primary human hepatocytes from liver tissue (TSP), described by *Baccarani et al* [46]

The biopsy material will be perfused under sterile conditions via a vessel (if available) of 25 ml of 2 mM Ethylenediaminetetraacetic acid (EDTA) solution (MilliporeSigma, USA) by disposable syringe and digested with a solution supplemented with 500 mg/L collagenase type P (MilliporeSigma, USA) at 37˚C. Washing in EDTA solution and tumor tissue digestion will be carried out on incubation shaker Celltron (Infors HT, Switzerland) at a constant speed of 60 min$^{-1}$ at 37˚C.

After that the fragment of the tumor will be carefully disrupted with a sterile scalpel, and the digested parenchyma will be collected in a test tube containing 50 ml of ice-cold RPMI 1640 (MilliporeSigma, USA) supplemented with 10% human serum (MilliporeSigma, USA). Three 1 mm$^3$ neoplasm fragments will be collected to create HCC-patient-derived xenograft- (PDX) avatars. The cell suspension will be filtered through 250-μm tissue strainers (Thermo Fisher Scientific, USA), and then washed three times at 50g for 5 minutes at 4˚C to remove cell debris.

Cell pellet will be resuspended in 4 ml of RPMI 1640 supplemented with 10% human serum and brought to final volume of 10 ml with the abovementioned medium.

### 1.2 Four-stage protocol for isolation of primary human hepatocytes from liver tissue (FSP), described by *Baccarani et al* [46]

The tumor tissue will be quickly washed from a disposable syringe at 37˚C under sterile conditions via a vessel (if available) with 25 ml of reperfusion solution, composed of: RPMI 1640 (Thermo Fisher Scientific, USA) with the addition of 150 mg/L glycine (MilliporeSigma, USA), 178 mg/L L-alanine (MilliporeSigma, USA), 40 IU insulin from bovine pancreas (MilliporeSigma, USA) and 1800 mg/L fructose (MilliporeSigma, USA). Three 1 mm$^3$ neoplasm fragments will be collected to create HCC-PDX-avatars.

After that, 25 ml of 2 mM EDTA solution (37˚C) will be perfused with a disposable syringe through a vessel (if available) to remove residual blood. Then the tumor tissue will be washed with 50 ml RPMI 1640 supplemented with 300 mg/L $CaCl_2$ using incubation shaker Celltron at a constant speed of 60 min$^{-1}$ at 37˚C.

Enzymatic digestion will be performed in a solution supplemented with 500 mg/L collagenase type P (MilliporeSigma, USA) with addition of 300 mg/L $CaCl_2$ using incubation shaker Celltron, 60 min$^{-1}$ at 37˚C. When the digestion of the parenchyma can be assessed visually, the tumor tissue will be washed with 50 ml of the reperfusion solution at 60 min$^{-1}$ at 37˚C.

After that, the tumor tissue will be disrupted with a sterile scalpel, and the cells will be collected after adding 50 ml of culture medium (RPMI 1640 + 10% human decomplemented plasma + 40 UI of insulin + 20 mL of 5% fructose) and filtration through 250-μm tissue strainers (Thermo Fisher Scientific, USA). The cell suspension will be washed three times at 50g for 5 minutes at 4˚C to remove cell debris.

Cell pellet will be resuspended in 4 ml of culture medium and brought to final volume of 10 ml with the abovementioned medium.

### 1.3 Protocol for isolation of primary human hepatocytes from diseased liver tissue (NDP), described by *Bhogal et al* [47]

The biopsy material will be washed with 1x phosphate buffered saline (PBS, pH 7,2, MilliporeSigma, USA) to identify suitable vessels that could be used for subsequent perfusion of buffers. All buffers used in the extraction procedure will be pre-warmed to 42˚C in a water bath. At this stage, the tumor tissue will be washed through the vessels with a 10 mM HEPES, pH 7,2 (MilliporeSigma, USA) to remove the remaining blood. Three 1 mm$^3$ neoplasm fragments will be collected to create HCC-PDX-avatars.

After that, the biopsy material will be impregnated with a chelating solution: 10 mM HEPES supplemented with 190 mg/L ethylene glycol bis (2-aminoethyl ether)-N,N,N′,N′-tetraacetic acid (EGTA), pH 7,2 (MilliporeSigma, USA), in order to disrupt the adhesion of cells to the underlying matrix. The process will be controlled visually. Then the neoplasm fragment will be washed with 10 mM HEPES (pH 7.2) to remove residual EGTA.

After that, the tissue fragment will be placed into enzyme buffer solution (75 ml), composed of: fresh aliquots of enzymes dissolved in Hank's balanced salt solution (HBSS) supplemented with 555 mg/L $CaCl_2$ and 475 mg/L $MgCl_2$ (Thermo Fisher Scientific, USA). After dissolution, the enzymes will be filtered through a 0.2-μm sterile syringe filter (Corning Inc., USA) back to HBSS. Enzyme composition: 0.5% w/v Collagenase A (MilliporeSigma, USA), 0.25% w/v Protease (MilliporeSigma, USA), 0.125% w/v Hyaluronidase (MilliporeSigma, USA) and 0.05% w/v Deoxyribonuclease (MilliporeSigma, USA). The digestion in the enzyme buffer will be conducted for 1–20 minutes. The digestion rate will be evaluated by palpation. Digestion will stop when the tumor tissue softens so much that it can be easily disrupted.

After that, the neoplasm tissue will be transferred to a sterile dish and disrupted with sterile scalpel in DMEM supplemented with 10% heat-inactivated fetal bovine serum (FBS, Thermo

Fisher Scientific, USA), 292 mg/L glutamine (Thermo Fisher Scientific, USA), 20 units/ml Penicillin G (MilliporeSigma, USA), 20 mg/L Streptomycin (MilliporeSigma, USA) and 2.5 mg/L Gentamycin. After dissociation of the fragment, the suspension will be passed through 250-μm tissue strainers (Thermo Fisher Scientific, USA), and then through 70-μm sterile nylon filter (Corning Inc., USA). Then the suspensions will be washed three times at 50g for 10 minutes at 4˚C in the culture medium.

Cell pellet will be resuspended in 4 ml of supplemented DMEM and brought to final volume of 10 ml with the abovementioned medium.

### 1.4 Protocol for isolation of primary human HCC cells (HCCP), described by *Cheung et al* [48]

The biopsy material will be washed with an Advanced MEM (AMEM, Thermo Fisher Scientific, USA) supplemented with 50 units/ml Penicillin G (MilliporeSigma, USA) and 50 mg/L Streptomycin (MilliporeSigma, USA), and then disrupted into 1 $mm^3$ fragments with a sterile scalpel. Three 1 $mm^3$ neoplasm fragments will be collected to create HCC-PDX-avatars.

After enzymatic dissociation with collagenase type IV (MilliporeSigma, USA) for 5 minutes at 37˚C three times, the disaggregated cell suspension will be filtered through a 40-μm sterile cell strainer (Corning Inc., USA).

Red blood cells will be lysed using the Red Blood Cell Lysis Buffer (MilliporeSigma, USA) and cells will be washed three times at 50g for 10 minutes at 4˚C in AMEM.

Cell pellet will be resuspended in 4 ml of AMEM and brought to final volume of 10 ml with the abovementioned medium.

### 2. Increasing the number of viable cells (described by Bhogal et al [47])

Cells yield and viability will be determined using the Cell Vitality Assay Kit, C12 Resazurin/ SYTOX™ Green (L34951, Thermo Fisher Scientific, USA) exclusion test using LUNA-II Automated Cell Counter (Logos Biosystems, South Korea).

If the cell viability is low (< 50%), but the resulting number of cells is high, the method of Percoll density gradient separation centrifugation will be used. Percoll (MilliporeSigma, USA) will be prepared by adding PBS (pH 7,2) and bringing the density to 4.5 g/ml. After that, Percoll will be added to the cells and suspension will be centrifuged at 300g for 30 minutes at room temperature. Then the cell viability will be determined again.

After counting, the HCC cell suspensions will be transferred to 25-$cm^2$ culture flasks (Corning Inc., USA), rotate gently and incubated in a humidified atmosphere containing 5% $CO_2$ overnight. Following overnight attachment, the medium will be decanted and replaced with a fresh preheated (i. e. 37˚C) medium.

### 3. Primary HCC cells cultivation

TSP- and FSP-isolated cells will be cultured according to protocol for cultivation of primary human hepatocytes from liver tissue.

NDP- isolated cells will be cultured according to protocol for cultivation of primary human HCC cells on a Poly-D-Lysine matrix.

HCCP-isolated cells will be cultured according to protocol for cultivation of primary human HCC cells on a gelatin matrix.

HepG2 cell line (CLS Cell Lines Service, Germany) will be used as HBV-negative control at all stages of our study.

### 3.1 Protocol for cultivation of primary human hepatocytes from liver tissue, described by *Baccarani et al* [46]

Cells will be resuspended in RPMI 1640 (MilliporeSigma, USA) supplemented with 10 mg/ L bovine insulin (MilliporeSigma, USA), 5 mg/L transferrin human (MilliporeSigma, USA), 0.019 mg/L somatostatin (MilliporeSigma, USA), 0.01 mg/L Gly-His-Lys acetate salt (MilliporeSigma, USA) and 10 nmol/L hydrocortisone acetate (MilliporeSigma, USA). Cell suspension will be placed in a 6-well plate (Corning Inc., USA) with seeding density $5 \times 10^5$ cells per well and cultured at 37°C in a humidified atmosphere containing 5% $CO_2$ with 12th-hour monitoring points for the formation of a monolayer.

### 3.2 Protocol for cultivation of primary human HCC cells on a Poly-D-Lysine matrix, described by *Qiu et al* [49]

Isolated cells will be resuspended in RPMI 1640 (Thermo Fisher Scientific, USA) supplemented with 10% FBS (Thermo Fisher Scientific, USA), 110 mg/L sodium pyruvate (MilliporeSigma, USA), 10 mg/L bovine insulin (MilliporeSigma, USA), 5.5 mg/L transferrin human (MilliporeSigma, USA), 40 ng/ml Epidermal Growth Factor (EGF) (MilliporeSigma, USA), 6.7 ng/mL sodium selenite (MilliporeSigma, USA) and placed into a 6-well plate (Corning Inc., USA) coated with 50 mg/L Poly-D-Lysine (Thermo Fisher Scientific, USA) with seeding density $5 \times 10^5$ cells per well. Cells will be cultured at 37°C in a humidified atmosphere containing 5% $CO_2$ with 12-hour control points for the formation of a monolayer.

### 3.3 Protocol for cultivation of primary human HCC cells on a gelatin matrix, described by *Cheung et al* [48]

Cells will be resuspended in Hepatocyte culture medium (Corning Inc., USA) and placed in 6-well plates coated with 0.1% gelatin (MilliporeSigma, USA) with seeding density $5 \times 10^5$ cells per well. The cell adhesion to the gelatin substrate is monitored for 12 hours.

### 3.4 HepG2 cells cultivation protocol is performed as described by *Donato et al* [50]

Quantitative determination of HBV DNA in the cultural supernatants will be performed 2 days after the cell culturing start by "AmpliSens® HBV Monitor-FRT" kit (Central Research Institute for Epidemiology of the Federal Service for Supervision of Consumer Rights Protection and Human Welfare, Russian Federation) using QuantStudio 5 real-time PCR system (Thermo Fisher Scientific, USA) for all cultivation protocols.

When cells are 80% confluent, they will be dissociated from the well surface using 0.05% trypsin-EDTA solution (MilliporeSigma, USA) and their viability will be evaluated by trypan blue exclusion test using LUNA-II Automated Cell Counter.

### *4. HCC-PDX-avatars development*

The establishment of HCC-PDX-avatars at first is caused by the need to adapt the mouse to human tissues for prevention an active "graft versus host" reaction on HCC-HBV-PDOX avatars stage later.

All animals will be housed in specific pathogen-free conditions in bioexclusion systems IsoCage N and IsoCage P (Tecniplast, Italy). All manipulations with animals are approved by Directive 2010/63/EU of the European Parliament and of the Council of 22 September 2010 on the protection of animals used for scientific purposes (22 September 2010), European Convention for the Protection of Vertebrate Animals Used for Experimental and Other Scientific Purposes (Strasbourg, 18 March, 1986) and Central Research Institute of Epidemiology Commission for the Care and Scientific Purposes Use of Animals (CRIECCSPUA, petition № 3-Zh). All surgical manipulations with animals will be carried out under anesthesia, and every effort will be made to minimize suffering.

5–6-week-old male BALB/c athymic nude mice (Charles River, Germany) will be randomly divided into six groups of 7 and subcutaneously injected into the left flank with one 1 mm$^3$ neoplasm fragment obtained during the primary HCC cells isolation protocols implementation:

- group # 1—TSP;

- group # 2—NDP;

- group # 3—HCCP.

All neoplasm fragments will be grafted in a mixture with 200 μL of ECM Gel from Engelbreth-Holm-Swarm murine sarcoma (MilliporeSigma, USA). The injection will be performed under Isoflurane (Laboratorios Karizoo, Spain) anesthesia (Biosthesia 300, Vilber Lourmat, France) using a syringe with 18G needle.

Group # 4 mice will be previously total body irradiated [51, 52] with 2 Gy by Xstrahl CIX3 irradiator cabinet (Xstrahl, UK) and injected in the same way with the neoplasm fragments obtained during the FSP implementation. During the irradiation procedure, the mice will be anesthetized with Isoflurane. Thus, we plan to evaluate the effect of pre-irradiation on tumor tissue engraftment and growth and on the possibility of HBV replication in pre-irradiated mice.

Group # 5 mice will be pre-irradiated with 2 Gy by Xstrahl CIX3 irradiator cabinet under Isoflurane anesthesia. Group # 5 and group # 6 mice will be subcutaneously injected into the left flank with $1 \times 10^7$/200 μL HepG2 cells resuspended in ECM Gel from Engelbreth-Holm-Swarm murine sarcoma [35, 53] and used as HBV-negative control.

Humane endpoints will be tumor size reaching of 2 cm in diameter or a body weight loss of at least 10%. Animal monitoring is carried out for no more than two months.

Weekly, until the condition requiring humane euthanasia of mice will be reached, the AFP (by Human alpha Fetoprotein ELISA Kit, Abcam, UK), HBsAg (by "HBsAg-IFA-BEST" kit, Vector-BEST, Russian Federation) and HBV DNA (by "AmpliSens® HBV Monitor-FRT" kit) levels in the blood serum will be assessed [48, 54]; mice weight, size of subcutaneous neoplasm, biochemical blood analysis (ALT, AST, TB, CB, AP, GGT, Chol) and complete blood count by auto hematology analyzer BC-2800Vet (Mindray Medical International Ltd, China) will be also performed weekly. ELISA tests will be performed using Multiskan FC microplate photometer (Thermo Fisher Scientific, USA). Blood samples from animals will be collected under Isoflurane anesthesia by the facial vein puncture with a 20G needle. The size of subcutaneous neoplasm will be measured using a micrometer screw gauge, and its volume will be calculated by the formula: tumor volume = (length×width$^2$)/2 [55]. When the tumors will be reached 0.4 to 0.6 cm in diameter, the tumor-bearing mice will be subjected to *in vivo* imaging studies.

*In vivo* imaging will be carried out by targeted delivery of fluorescently labeled antibodies to the corresponding targets. Before labeling antibodies, a Microcon-10kDa centrifugal filter units with ultracel-10 membrane (MilliporeSigma, USA) will be used to remove residual sodium azide. Antibody labeling will be performed in accordance with the protocol provided by the manufacturer. *In vivo* HBsAg fluorescent intensity imaging will be implemented on small animal optical imaging system Newton 7.0 FT-500 (Vilber Lourmat, France) with F-550 filter and "c 540 nm" lighting using the Hepatitis B Virus Surface Monoclonal Antibody (Thermo Fisher Scientific, USA) labeled with Mix-n-Stain CF 555 Antibody Labeling Kit (MilliporeSigma, USA) [56–58]. Xenograft growth estimation will be carried out on the Newton 7.0 FT-500 by means of *in vivo* fluorescent intensity imaging of AFP Monoclonal Antibody (SP154) (Thermo Fisher Scientific, USA) labeled with SAIVI Rapid Antibody Labeling Kit, Alexa Fluor 680 (Thermo Fisher Scientific, USA) using the Evolution-Capt Edge software

(Vilber Lourmat, France) with F-700 filter and "c 680 nm" lightning. The concentration of injected antibodies will be selected empirically, but the total volume of the injected solution would not exceed 0.2 ml per mouse. Labeled HBV surface monoclonal antibodies will be injected into the tail vein using sterile disposable syringe with 29G needle. AFP monoclonal antibodies will be delivered in the same way 1 hour later.

Upon reaching the humane endpoint, animals will be sacrificed by total exsanguination under anesthesia (Isoflurane inhalation; Zoletil (Virbac, France) + Xyla (Nita-Farm, Russian Federation), 7.5–50 mg/kg and 5–10 mg/kg, respectively, intraperitoneally). Animals from the control groups will be sacrificed on the same day.

Blood samples, subcutaneous xenografts, spleen, liver and tissue metastases (if available) will be collected for further routine histological hematoxylin-eosin (H&E) staining and IHC.

1 mm$^3$ subcutaneous xenograft tissues fragments obtained from mice with the greatest ability of HBV DNA replication will be separated under sterile conditions for further cells cultivation and HCC-HBV-PDOX-avatars generation.

### 5. Isolation and cultivation of primary HCC cells obtained from HCC-PDX-avatars

The isolation and cultivation of primary cell culture obtained from subcutaneous xenografts of HCC-PDX-avatars will be carried out according to the protocol proposed by the Research laboratory of Xin Chen [59].

If the cell viability is low (< 50%), but the resulting number of cells is high, the method of Percoll density gradient separation centrifugation will be used as described in the "Increasing the number of viable cells" section.

Quantitative determination of HBV DNA in the culture supernatant will be performed 2 days after the cell culturing start by the "AmpliSens® HBV Monitor-FRT" kit using QuantStudio 5.

When cells are 80% confluent, they will be dissociated from the well surface using 0.05% trypsin-EDTA solution and their viability will be evaluated by trypan blue exclusion test using LUNA-II Automated Cell Counter.

### 6. HCC-HBV-PDOX-avatars development

5–6-week-old male BALB/c athymic nude mice will be randomly divided into groups, with 7 mice in each group. If high results will be demonstrated in group # 5 of HCC-PDX-avatars, the animals are divided into 4 groups:

- group #1 mice will be grafted by 1 mm$^3$ tumor fragments obtained from HCC-PDX-avatars with the highest level of HBV DNA replication;

- group # 2 mice will be grafted by 1 mm$^3$ tumor fragments obtained from group # 6 of HCC-PDX-avatars and used as HBV-negative control for group #1 of HCC-HBV-PDOX-avatars;

- group #3 mice will be pre-irradiated with 2 Gy by Xstrahl CIX3 irradiator cabinet under Isoflurane anesthesia and grafted by 1 mm$^3$ tumor fragments obtained from group # 4 of HCC-PDX-avatars;

- group # 4 mice will be pre-irradiated with 2 Gy by Xstrahl CIX3 irradiator cabinet under Isoflurane anesthesia and grafted by 1 mm$^3$ tumor fragments obtained from group # 5 of HCC-PDX-avatars and used as HBV-negative control for group #3 of HCC-HBV-PDOX-avatars.

If the level of HBV DNA replication in pre-irradiated HCC-PDX-avatars will be low, the study will involve only groups # 1 and # 2.

All surgical procedures will be carried out under sterile conditions. Laparotomy will be performed on mice under anesthesia (Isoflurane inhalation; Zoletil + Xyla, 7.5–50 mg/kg and 5–10 mg/kg, respectively, intraperitoneally) to the left of the *linea alba*. The fibrous capsule and the parenchyma of the left lobe of the liver will be perforated to 3 mm depth with an aspiration 18G biopsy needle, and a fragment of the liver parenchyma will be extracted. A 1 mm$^3$ subcutaneous neoplasm fragment derived from the HCC-PDX-avatar will be placed in the formed cavity. Similar manipulations will be performed with the spleens of 4 mice in each group. The liver and spleen wounds will be closed with Surgicel (Johnson & Johnson, USA) [55, 60, 61]. Muscle wounds will be sutured continuously with Vicryl W9113 4/0. Skin wounds will be sutured continuously with Ethilon 4/0. Skin sutures will be treated with Terramycin aerosol spray (Zoetis Inc, USA) once. After surgical procedures animals will be treated with Ketoprofen 1% solution (Merial, France) subcutaneous injection, 2 mg/kg, once a day, 1–3 days.

AFP, HBsAg and HBV DNA levels, biochemical blood analysis, complete blood count, HBsAg *in vivo* fluorescent intensity imaging, mice weight and xenograft growth will be estimated as described in the "HCC-PDX-avatars development" section.

Humane endpoints and the method of euthanasia will be the same as described in the "HCC-PDX-avatars development" section.

*7. Morphology studies design*

It is planned to perform macro-and micromorphological evaluations.

Macromorphological evaluation will be performed during the necropsy process.

Herewith the size and mass of the xenograft, liver and spleen, as well as other pathological changes (if available) will be determined.

Formalin fixation and paraffin embedding of tissues will be performed according to the standard procedure [62].

The paraffin-embedded tissues sectioning will be carried out by rotary microtome Rotary 3003 (PFM medical, Germany).

Micromorphological evaluation will be performed according to the routine H&E staining technique of paraffin-embedded tissue sections [62].

IHC staining of paraffin-embedded tissue sections will be performed according to the modified protocol described by *Liu et al* [39]:

- Tissue blocks will be cut, mounted on microscope slides and heated at a temperature of 56 ˚C for 20 minutes in a dry oven.

- Paraffin will be removed with xylene, and the tissues are consistently rehydrated by reducing the concentration of ethanol (100%, 90%, 70%) to deionized water.

- Antigen will be extracted using the pre-treatment module for tissue specimens PT Link (Dako, Agilent Technologies, USA) using a 10 mM sodium citrate buffer (pH 6,0).

- Slides will be washed 3 times in 1xPBS for 5 minutes with gentle agitation.

- Tissue samples will be blocked for 1 hour with PBS supplemented with 10% goat serum donor herd (G6767-100ML, MilliporeSigma, USA) at room temperature in a light-protected chamber.

- Slides will be washed twice in 1xPBS and once in PBS-1% Tween-20 for 15 minutes with gentle agitation.

- Tissue samples will be treated with the Mouse on Mouse Polymer IHC Kit (ab269452, Abcam, UK) for 30–60 minutes at room temperature in a light-protected camera.

- Slides will be washed twice in 1xPBS and once in PBS-1% Tween-20 for 15 minutes with gentle agitation.

- Slides will be incubated with specific primary antibodies in a light-protected chamber at 4 ˚C overnight: HBcAg monoclonal antibody (MA1-7607, Thermo Fisher Scientific, USA) and AFP rabbit polyclonal antibody (PA5-16658, Thermo Fisher Scientific, USA) resuspended in blocking buffer.

- Slides will be washed 3 times in 1xPBS for 5 minutes with gentle agitation.

- After that, slides will be stained with corresponding secondary antibodies for 1 hour at room temperature in a light-protected chamber: goat anti-mouse IgG (H+L) highly cross-adsorbed secondary antibody, Alexa Fluor 488 (A-11029, Thermo Fisher Scientific, USA) and goat anti-rabbit IgG (H+L) cross-adsorbed secondary antibody, PE (P-2771MP, Thermo Fisher Scientific, USA) resuspended in blocking buffer.

- Slides will be washed 3 times in 1xPBS for 5 minutes with gentle agitation.

- Nuclei will be counterstained using DAPI (D9542-10 MG, MilliporeSigma, USA).

- Slides will be washed 3 times in 1xPBS for 5 minutes with gentle agitation.

- Slides will be dried for 1–2 hours at room temperature in a light-protected chamber.

Slides will be examined using an inverted research microscope Olympus IX73 (Olympus Corporation, Japan).

*8. Data analysis*

Data analysis and graphing will be performed using Mann-Whitney U test, Kruskal-Wallis H test and three-way analysis of variance (ANOVA) by Prism 9 (GraphPad Software, USA).

Data fluorescent intensity images will be performed using Kuant software (Vilber Lourmat, France).

## Discussion

High levels of HBV DNA in the blood serum of patients correlate in clinical conditions with the accumulation of liver damage, resulting to cirrhosis and HCC development [25]. Therefore, the simulation of the clinical features of HBV infection requires animal models to adequately display the disease course. The methods of creating HCC-HBV-PDOX-avatars made it possible to conduct *in vivo* studies, including therapeutic protocols, directly on models as close to the original as possible. Therefore, we set ourselves the goal to evaluate the effectiveness of the protocols for isolation and cultivation of primary HCC cells described in the literature and the methods for introducing xenografts into the animal according to the following criteria:

- number of isolated viable primary HCC cells;

- viability of primary HCC cells during cultivation;

- efficiency of *in vitro* HBV DNA replication;

- effectiveness of xenograft engraftment in the mouse;

- effectiveness of *in vivo* HBV DNA replication;

- correlation between the results of measurements of AFP levels, biochemical blood analysis (ALT, AST, TB, CB, AP, GGT, Chol) and complete blood count obtained during the study with similar data obtained from donors.

The primary HCC cells isolation protocol will be considered acceptable if the number of viable HCC cells is not less than 50%.

The primary HCC cells viability will be considered acceptable if the cell's monolayer is achieved within 5 days in a 6-well plate.

The efficiency of *in vitro* HBV replication will be considered acceptable if the amount of DNA will reach $10\pm7\times10^5$ viral copies/mL of the supernatant during the cultivation period [63].

The effectiveness of xenograft engraftment in the mouse will be considered acceptable if the size of the tumor increases during the period of endpoints reaching.

The efficiency of *in vivo* HBV DNA replication will be considered acceptable if the level of HBV DNA in blood reaches $10\pm6\times10^5$ viral copies/mL [64].

Our proposed model differs from the existing ones by the low invasiveness of surgical manipulations with animals. In addition, it is expected that X-ray pre-irradiation of the total mouse body and step-by-step transplantation will improve engraftability of the tumor and reduce the "graft versus host" reaction.

Our study will be the first extensive comparative study of popular modern methods and protocols for the isolation and cultivation of primary HCC cells and the establishment of HCC-HBV-PDOX-avatars for the study of chronic HBV infection. It will be the first large-scale study of *in vivo* AFP and HBsAg distribution and localization, performed on the small animal optical imaging system Newton 7.0 FT-500 and confirmed by images. All protocols will be optimized and characterized. Some methods will be considered as "research-use only", others will be recommended for use in translational biomedical research.

## Author Contributions

**Conceptualization:** Aleksey M. Nagornykh, Marina A. Tyumentseva, Aleksandr I. Tyumentsev.

**Data curation:** Aleksey M. Nagornykh, Marina A. Tyumentseva, Aleksandr I. Tyumentsev.

**Formal analysis:** Aleksey M. Nagornykh, Marina A. Tyumentseva, Aleksandr I. Tyumentsev.

**Funding acquisition:** Vasiliy G. Akimkin.

**Investigation:** Aleksey M. Nagornykh, Marina A. Tyumentseva, Aleksandr I. Tyumentsev.

**Methodology:** Aleksey M. Nagornykh, Marina A. Tyumentseva, Aleksandr I. Tyumentsev.

**Project administration:** Aleksey M. Nagornykh, Marina A. Tyumentseva, Aleksandr I. Tyumentsev.

**Supervision:** Vasiliy G. Akimkin.

**Writing – original draft:** Aleksey M. Nagornykh, Marina A. Tyumentseva, Aleksandr I. Tyumentsev.

**Writing – review & editing:** Vasiliy G. Akimkin.

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
