## [Decision Letter · Decision Letter 0]

7 Oct 2021

PONE-D-21-25364Protocol for chronic hepatitis B virus infection mouse model development by patient-derived orthotopic xenograftsPLOS ONE

Dear Dr. Nagornykh,

Thank you for submitting your manuscript to PLOS ONE. After careful consideration, we feel that it has merit but does not fully meet PLOS ONE’s publication criteria as it currently stands. Therefore, we invite you to submit a revised version of the manuscript that addresses the points raised during the review process.

We look forward to receiving your revised manuscript.

Kind regards,

Ezio Laconi, MD, PhD

Academic Editor

PLOS ONE

“This work is supported by the Ministry of Science and Higher Education of the Russian Federation within the framework of a grant in the form of a subsidy for the creation and development of the «World-class Genomic Research Center for Ensuring Biological Safety and Technological Independence under the Federal Scientific and Technical Program for the Development of Genetic Technologies», agreement No. 075-15-2019-1666.”

We note that you have provided information within the Funding Section. Please note that funding information should not appear in other areas of your manuscript. We will only publish funding information present in the Funding Statement section of the online submission form.

 “This work is supported by the Ministry of Science and Higher Education of the Russian Federation within the framework of a grant in the form of a subsidy for the creation and development of the «World-class Genomic Research Center for Ensuring Biological Safety and Technological Independence under the Federal Scientific and Technical Program for the Development of Genetic Technologies», agreement No. 075-15-2019-1666.

The funders had and will not have a role in study design, data collection and analysis, decision to publish, or preparation of the manuscript.”

4. Please amend the manuscript submission data (via Edit Submission) to include author Vasiliy G. Akimkin

Reviewers' comments:

Reviewer's Responses to Questions

**Comments to the Author**

1. Does the manuscript provide a valid rationale for the proposed study, with clearly identified and justified research questions?

Reviewer #1: No

Reviewer #2: Partly

2. Is the protocol technically sound and planned in a manner that will lead to a meaningful outcome and allow testing the stated hypotheses?

Reviewer #1: Partly

Reviewer #2: No

3. Is the methodology feasible and described in sufficient detail to allow the work to be replicable?

Reviewer #1: No

Reviewer #2: No

4. Have the authors described where all data underlying the findings will be made available when the study is complete?

Reviewer #1: No

Reviewer #2: No

5. Is the manuscript presented in an intelligible fashion and written in standard English?

Reviewer #1: No

Reviewer #2: No

6. Review Comments to the Author

You may also provide optional suggestions and comments to authors that they might find helpful in planning their study.

Reviewer #1: Aleksey et al tried to compare and establish an optimized protocol to generate HBV-HCC PDX mouse model. However, the current version needs to be improved.

Major comment,

1. No any realy result is shown. Please provide your own data.

2. Authors should illustrate their technical protocol in figures instead of narrate their methods only in words.

3. Material and methods part is baddly organized.

Reviewer #2: The rationale and objectives should be presented in separate paragraphs with titles. The abstract lacks a section on results, the format is truncated.

Reference 5 is not a trimera mouse model, should be removed as it does not pertain to the current subject.

Description of the humanized mice should include a review on the models available for HBV research.

Page 3 the authors state « The currently existing in vitro and in vivo HBV models make it possible to observe … through bioluminescence and fluorescence… allowing to assess the persistence and replication of viral DNA… do not require periodic euthanasia of the animal. » what is the litterature showing this? none to my knowledge, please cite the litterature.

Study methods section: « d) q » what does this mean?

What is the rational for irradiating mice after the grafts?

Page 8 the authors state « HBsAg in vivo visualisation is performed on Vilber Newton 7.0 … is used as a delivery agent » . what is the litterature showing this? none to my knowledge, please cite the litterature. The cited reference 41 only has in vitro data.

Animal ethics need to be better described and implemented: mice that undergo surgical procedures need to be treated with analgesic, not only anaesthetic, and humane endpoints of the maximum tumor size limits and weight loss should be included in the protocol description.

7. PLOS authors have the option to publish the peer review history of their article (what does this mean?). If published, this will include your full peer review and any attached files.

Reviewer #1: No

Reviewer #2: No

---

## [Author Response · Author response to Decision Letter 0]

19 Nov 2021

Dear Dr. Laconi!

We have tried to adjust our manuscript, taking into account all your comments, in accordance with the requirements of PLOS ONE. We would like to inform you that we have deposited our protocol in protocols.io and got a DOI: dx.doi.org/10.17504/protocols.io.bz7sp9ne

Best regards,

Aleksey Nagornykh.

Response to Reviewer #1 comments:

Dear Reviewer,

We want to thank you for the thoughtful feedback. We have incorporated all of your comments to our revision and added some points to discussion. We also provide a point-by-point reply to all of your comments.

Comment 1:

As is mentioned in https://journals.plos.org/plosone/s/what-we-publish#loc-study-protocols Study Protocols describe detailed plans for conducting research, including the background, rationale, objectives, methodology, statistical plan, and organization of a research project.

Comment 2:

We’ve supplemented the manuscript with graphics pipelines.

Comment 3:

We tried to reorganize the “Material and methods” section.

On behalf of authors, I want to thank Reviewer #1. All comments were very helpful.

Best regards,

Aleksey Nagornykh.

Response to Reviewer #2 comments:

Dear Reviewer,

On behalf of all co-authors, I thank you for your time and efforts on revising the manuscript. We now submit the revised text. We’ve carefully revised our manuscript according to your suggestions.

Comment 1:

We’ve adjusted the sections “Research objectives” and “Study aim”. Expected results are described in the “Discussion” section. No data sets were generated or analyzed during the current study. All relevant data of this study will be made available upon study completion. Results will be presented in one of the peer-reviewed specialized journals in accordance with the protocol described in our manuscript.

Comment 2:

We’ve changed Reference 5 to a more suitable one.

Comment 3:

We’ve included description of the humanized mice to “Introduction” section. 

Comment 4:

That was a typo. Models of viral infections were implied. We’ve included information about the literature to “Introduction” section, references 40-44.

Comment 5:

That was a typo. qPCR was implied.

Comment 6:

We don’t plan to irradiate mice after the grafts. There is no mention of irradiating mice after grafts in our manuscript.

Comment 7:

We’ve included examples of literature to “HCC-PDX-avatars development” section, references 56, 57.

Comment 8:

We’ve supplemented “HCC-HBV-PDOX-avatars development” section with information about postoperative animal care. Humane endpoints were defined in the first version of the manuscript: “Observation of the animals is continued until the tumor reaches of 2 cm in diameter or a body weight loss of more than 10%, but not more than 2 months”. When working on the manuscript after your comment, this phrase was reformulated.

On behalf of authors, I want to thank Reviewer #2. All comments were very helpful.

Best regards,

Aleksey Nagornykh.

---

## [Editor Report · Decision Letter 1]

8 Feb 2022

Protocol for chronic hepatitis B virus infection mouse model development by patient-derived orthotopic xenografts

PONE-D-21-25364R1

Dear Dr. Nagornykh,

We’re pleased to inform you that your manuscript has been judged scientifically suitable for publication and will be formally accepted for publication once it meets all outstanding technical requirements.

Kind regards,

Md. Golzar Hossain, Ph.D.

Academic Editor

PLOS ONE
---

## [Editor Report · Acceptance letter]

11 Feb 2022

PONE-D-21-25364R1 

Protocol for chronic hepatitis B virus infection mouse model development by patient-derived orthotopic xenografts 

Dear Dr. Nagornykh:

I'm pleased to inform you that your manuscript has been deemed suitable for publication in PLOS ONE. Congratulations! Your manuscript is now with our production department. 

Kind regards, 

on behalf of

Dr. Md. Golzar Hossain 

Academic Editor

PLOS ONE